# BCSCN:Reducing Domain Gap through Bézier Curve basis-based Sparse Coding Network for Single-Image Super-Resolution

## ABSTRACT

Single Image Super-Resolution (SISR) is a pivotal challenge in computer vision, aiming to restore high-resolution (HR) images from their low-resolution (LR) counterparts. The presence of diverse degradation kernels creates a significant domain gap, limiting the effective generalization of models in real-world scenarios. This study introduces the Bézier Curve basis-based Sparse Coding Network (BCSCN), a preprocessing network designed to mitigate input distribution discrepancies between the training and testing phases of super-resolution networks. BCSCN achieves this by removing visual defects associated with the degradation kernel in LR images, such as artifacts, residual structures, and noise. Additionally, we propose a set of rewards to guide the search for basis coefficients in BCSCN, enhancing the preservation of main content while eliminating information related to degradation. The experimental results highlight the importance of BCSCN, showcasing its capacity to effectively reduce domain gaps and enhance the generalization of super-resolution networks. Upon acceptance, we will make our code and models publicly available.

## CCS CONCEPTS

• **Computing methodologies** → **Reconstruction**.

## KEYWORDS

Super-Resolution, Sparse Coding, Reinforcement Learning

## 1 INTRODUCTION

Single Image Super-Resolution (SISR) is a fundamental problem in computer vision, with the goal of reconstructing high-resolution (HR) images from their corresponding low-resolution (LR) counterparts. In response to the surge in deep learning, approaches leveraging deep neural networks have garnered significant attention for addressing image super-resolution (SR) tasks. These methods typically rely on collected HR-LR image pairs for training [3, 12, 15, 17, 23, 24, 34, 43]. However, obtaining such pairs in real-world scenarios can be arduous, often requiring the generation of LR images from HR originals. Most SR techniques resort to simplistic bicubic downsampling to create synthetic HR-LR pairs, but this methodology introduces a significant challenge: the **domain gap** emerges due to the disparity in the distribution of LR images between the training and testing phases, stemming from the use

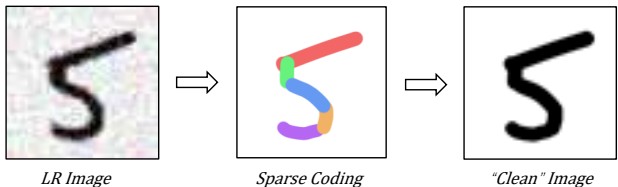

*LR Image*  *Sparse Coding*  *"Clean" Image*

**Figure 1: The figure illustrates a schematic representation of the reconstruction process for transforming a severely degraded LR image into a "clean" image using Bézier curve basis functions. Different colors distinguish various Bézier curve bases, all of which have coefficients set to one. This reconstruction process effectively eliminates the blurred edges of the image while simultaneously addressing artifacts and chromatic aberrations introduced by degradation.**

of distinct degradation kernels [21, 32, 37]. This domain gap hinders the models' capacity to generalize effectively, impacting their practical utility in real-world applications where the degradation kernels are intricate and unknown, differing from those applied in LR image generation during training.

In most cases, three distinct visual defects emerge during the degradation process of an image: **Artifacts**: These are isolated artifacts generated during image compression or processing that are unrelated to the image content. **Residual structures**: These are isolated structures that form due to the severe loss of semantic contextual information during the image degradation process. **Noise**: This includes common Gaussian noise models and non-Gaussian noise caused by camera sensors. However, for different degradation kernels, such as the real degradation processused in LR image generation, these visual defects may manifest differently. This variation is the root cause of the domain gap.

This raises the question of whether there exists an image preprocessing operation, independent of the degradation kernel, capable of eliminating visual effects associated with the degradation kernel. These effects include artifacts, residual structures, and noise, while concurrently preserving content to alleviate the domain gap. The objective is to attain a "clean" image through this operation, one that is imperceptible in terms of degradation, yet retains abundant content and detail. Subsequent to this preprocessing step, a SR network with superior generalization performance can be trained using these "clean" images, effectively learning the mapping from "clean" images to HR images.

In this paper, to achieve this goal, we present a preprocessing neural network, the Bézier Curve basis-based Sparse Coding Network (BCSCN), designed to generate the "clean" image. BCSCN integrates an over-complete Bézier curve basis space, representing degraded LR images through linear combinations of sparse Bézier curve basis functions. To determine the basis coefficients for a given

LR image, BCSCN employs a basis coefficients search agent. The ultimate "clean" image with sharp edges is reconstructed based on the LR image's bases and coefficients using a differentiable neural renderer within BCSCN.

The key innovation lies in utilizing a well-designed and restricted/sparse set of Bezier curve basis functions. This choice allows for the exclusion of details associated with the degradation kernel, such as artifacts, residual structures, and noise. Consequently, the final reconstruction not only preserves the essential image content but also effectively eliminates information related to the degradation, as illustrated in Figure 1.

Importantly, BCSCN is independent of the degradation kernel. It can be trained on samples generated by a single, simple degradation kernel (such as bicubic degradation kernels) and applied to LR images generated by other degradation kernels to obtain their corresponding "clean" images.

In summary, the primary contributions of this article can be outlined as follows:

- We introduce BCSCN, a preprocessing network based on the Bézier curve basis. BCSCN effectively minimizes input distribution discrepancies between training and testing phases of the subsequent SR network, thereby addressing domain gap challenge.
- We present a set of rewards designed to guide the basis coefficients search agent in capturing main content and eliminating information associated with degradation.
- BCSCN-ESRGAN and BCSCN-PASD$_b$, achieved by integrating BCSCN into GAN-based and diffusion-based SR networks, have significantly improved the naturalness performance on real-world datasets with unknown degradation kernels, compared to their respective base models. Additionally, BCSCN-PASD$_b$ has achieved results competitive to state-of-the-art (SOTA) blind SR methods while requiring fewer samples and lower training resources.

## 2 RELATED WORK

### 2.1 Single Image Super-Resolution

SRCNN [5] introduced a three-layer neural network and standed as a pioneer in SISR. Since then, researchers turned to focus on developing larger and deeper networks using residual architectures. For example, a 20-layer deep neural network was used by VDSR [12] based on residual learning. EDSR [17] took a step further, featuring a remarkably deep and wide network with modified residual blocks for SR tasks. RCAN [43] introduced channel attention and further designed a network with 1000 layers. However, convolutional sliding window mechanism limits the ability of a network to harness global contextual information. Hence, HAN [24] incorporated layer-wise, channel-wise, and spatial attention modules, capturing hierarchical features by considering inter-layer correlations. Recently, new methods have been developed to capture long-range relationships between pixels in images. This allows them to have larger receptive fields and achieve better PSNR performance [3, 15, 23]. Nevertheless, these methods exhibit sensitivity to distribution discrepancies in LR images, resulting in artifacts when inference distribution deviates from the training distribution, thereby severely constraining their real-world applicability.

### 2.2 Learning Degradation Process

In an effort to narrow the domain gap between training and real data samples, some researchers have explored the degradation process of images to align training data distribution with real-world test data distribution. Some have achieved this by adjusting camera focal lengths, directly capturing paired data under real-world conditions [1, 36]. Other researchers have used methods that rely on pre-defined degradation kernels. These methods synthesize degraded images by artificially creating complex degradation kernels that aim to cover the distribution of datasets from various real-world scenarios [18, 21, 31, 33, 40, 41]. To generate LR images that closely match specific real test image domains, some approaches [22, 32, 37] have opted for adaptive learning of degradation, leveraging GANs to explicitly or implicitly learn the degradation process from real target distributions during training, thus bringing training samples closer to specific target real data distributions.

we leverage the robustness of sparse coding against the impact of degradation factors, effectively addressing distribution discrepancies between synthetic and real data, all without the need for prior knowledge of the specific degradation distribution. Our approach, trained solely on data synthesized using simple bicubic downsampling, significantly enhances model generalization performance.

### 2.3 Image Sparse Coding

Sparse coding has been extensively explored as a conventional approach in the field of computer vision. Previous methods [6, 14, 38, 39] typically involved learning an overcomplete dictionary, from a set of training images. During the reconstruction phase, the optimization process focuses on finding a set of coefficients that correspond to the atoms of the dictionary. These coefficients aim to yield the best linear combination to reconstruct the test images. However, due to the challenge of learning a sufficiently overcomplete dictionary, its reconstructive capabilities are limited when applied to natural images. In this paper, we define a sparse dictionary as a continuous basis space composed of second-order Bézier curves. We employ a reinforcement learning agent to explore sparse basis coefficients, enabling accurate reconstruction of LR images while alleviating degradation factors such as residual structures, noise, and artifacts.

## 3 METHOD

By introducing our proposed BCSCN, the super-resolution (SR) process is divided into two stages: the initial BCSCN preprocessing stage and the subsequent SR network stage, as illustrated in Figure 2(a). The BCSCN incrementally reconstructs the LR image $I$ into a "clean" image $\hat{I}_T$ by searching for the optimal basis coefficients within a predefined basis space. Subsequently, the resultant "clean" image $\hat{I}_T$ is fed into the SR network to produce the output HR image.

### 3.1 Definition of Basis Space

Given over-complete bases $\{\phi_i(\cdot)|i = 1, \cdots, N\}$, a degraded LR image $I$ can be represented by:

$$I = \sum_i \alpha_i \phi_i, \tag{1}$$

where $\alpha$ is the coefficient of the basis functions.

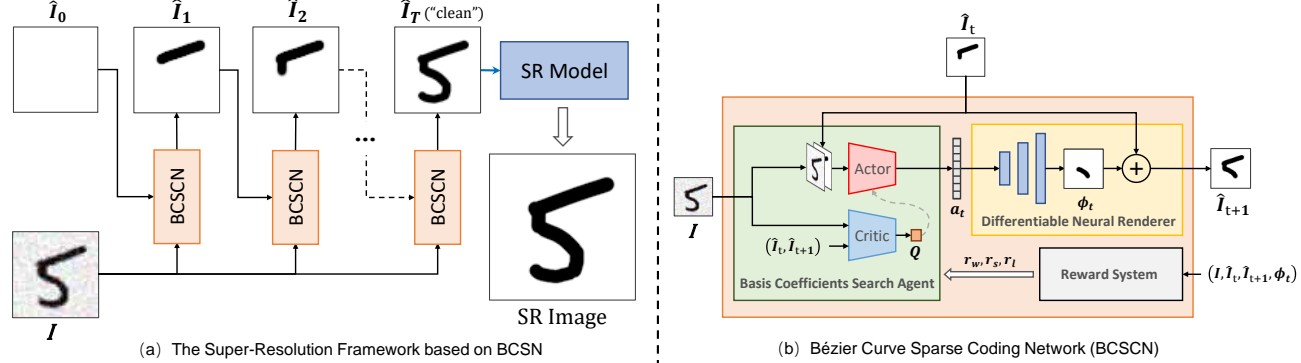

(a) The Super-Resolution Framework based on BCSN      (b) Bézier Curve Sparse Coding Network (BCSCN)

**Figure 2: Illustrating our proposed BCSCN-based super-resolution framework. a) The operational sequence of the BCSCN. BCSCN starts from $\hat{I}_0$, generating the next reconstruction image $\hat{I}_{t+1}$ based on the current reconstruction image $\hat{I}_t$ and LR image $I$. Through $T$ iterations, the "clean" image $\hat{I}_T$ is obtained, which is then fed into the SR network to produce a high-resolution output. b) Depicting the architecture of BCSCN, consisting of a coefficient-searching agent, differentiable neural renderer (DNR), and reward system. Guided by the reward system, the agent generates basis function parameters based on the current reconstruction image $\hat{I}_t$ and $I$. These parameters are overlaid with $\hat{I}_t$ through DNR to generate $\hat{I}_{t+1}$.**

In this paper, we opt for a space of Bézier curve bases with diverse attributes, including shape, width, and color, as the foundations. And a basis of this space is defined as:

$$\phi_i = \{\mathcal{B}_i, w_i, C_i\}, \qquad (2)$$

where $\mathcal{B}_i$ denotes the shape, defining as a second-order Bézier curve that is controlled by a set of specified control points $P_0^i, P_1^i, P_2^i$:

$$\mathcal{B}_i(P_0^i, P_1^i, P_2^i) = (1-t)^2 P_0^i + 2(1-t)t P_1^i + t^2 P_2^i, t \in [0,1] \qquad (3)$$

By flexibly adjusting the positions of these control points, various natural image elements such as points, lines, and curves with sharp edges can be formed.

And in Eq. 2, $w_i$ signifies the curve's width, and $C_i$ is a three-element tuple representing its color attributes $(r, g, b)$. This enhancement amplifies the expressive capacity of the Bézier curve, allowing for a better fit to natural shapes.

As discussed in Section 1, our approach involves employing a carefully designed and sparse set of bases, which proves effective in preserving edge details while eliminating defects introduced by degradations. The utilization of sparse codes for image representation, with a significant portion of coefficients $\alpha_i$ being zero, allows the model to focus on crucial features in the input image, thereby effectively removing defects associated with degradation.

To extract the main content from the image, a small number of bases are selected from the entire set to reconstruct a LR image. Sparsity control operates on two aspects: the area of the curves and the number of curve bases. By managing these aspects, the decomposed bases can effectively capture the main content of the image.

## 3.2 Basis Coefficients Search

In this paper, we enforce constraints on the basis coefficients to be either one or zero, preventing the combination of multiple bases from forming a transparent element — an unusual occurrence in real images. This transformation shifts the decomposition process

from calculating basis coefficients to a basis selection procedure, significantly reducing the complexity of our proposed method. However, it is important to note that this remains an NP-hard problem, and we address this challenge by utilizing reinforcement learning methods.

Furthermore, we model the base selection procedure as a sequential decision-making procedure, utilizing a base coefficient search agent to minimize the reconstruction error between the "clean" image and the LR image. The objective is to identify the optimal Bézier curve base representation within a fixed number of bases. LearningToPaint [10] has already confirmed the effectiveness of this method in efficiently reconstructing realistic and natural images in complex environments.

As depicted in Figure 2(b), given the LR image $I$ and the current reconstruction result $\hat{I}_t$, the agent's objective is to predict the parameters of the next basis $a_t$ based on $\hat{I}_t$ and $I$.

Next, we will delineate the state space and action space for the basis coefficient search agent, as well as the reward function.

*3.2.1 State & Action.* The state space is composed of three components: the LR image $I$, the current reconstruction image $\hat{I}_t$, and the current step $t$. It is represented as a triplet: $s_t = \{I, \hat{I}_t, t\}$.

Bézier curve bases are determined by a set of parameters, and the action space of the agent consists of these parameters, as depicted below:

$$a_t = \{P_0^t, P_1^t, P_2^t, w^t, r^t, g^t, b^t\}, \qquad (4)$$

where $\{P_0^t, P_1^t, P_2^t\}$ represent the control points of the shape of $\phi_t$, $w^t$ is the width of the second-order Bézier curve, and $(r^t, g^t, b^t)$ signifies the color attributes of the basis. We employ a pre-trained differentiable neural renderer [10], which takes the bases parameters $a_t$ as input and produces the pixel-space Bézier curve basis $\phi_t$ as output. Subsequently, $\phi_t$ is superimposed onto $I_t$ to yield the next reconstructed image $I_{t+1}$, resulting in the next state $s_{t+1} = \{I, \hat{I}_{t+1}, t+1\}$.

3.2.2 *Reward.* The agent's objective is to maximize cumulative rewards, aiming to ensure that each step's reconstruction operation is closer to $I$. To achieve this, we devised a tiered reward mechanism expressed as follows:

$$R(s_t, a_t) = r(s_{t+1}) - r(s_t), \tag{5}$$

where $R(s_t, a_t)$ represents the reward function at step $t$, and $r(s_t)$ is defined as the reward measure of the operation results $\hat{I}_t$ at step $t$ in relation to $I$.

Inspired by [10], we employed the WGAN with gradient penalty (WGAN-GP) [7] as the reconstruction distance measure between the current reconstruction image $\hat{I}_t$ and the target image $I$, specifically defined as:

$$r_w(s_t) = D(\hat{I}_t, I), \tag{6}$$

where $D(\cdot)$ is the discriminator.

Additionally, to encourage the agent to choose a basis which represents a larger curve area, ensuring sparsity, capturing prominent image features, and effectively mitigating degradation factors in image reconstruction, we introduced a curve area reward:

$$r_s(s_t) = \mathcal{S}(\phi_t), \tag{7}$$

where $\mathcal{S}(\cdot)$ is used to compute the area occupied by the curve represented by basis $\phi_t$ in the reconstructed image.

However, the experimental findings reveal that when $I$ is exclusively reconstructed using bases that represent large curves, it can lead to over-smoothing. This results in the preservation of only the main structures while fine details are sacrificed. Therefore, we further introduced high-frequency response reward to encourage agents to use bases with smaller curve in the areas with rich details, to preserve more details in the reconstruction results. This reward can be expressed as:

$$r_l(s_t) = \text{Laplace}(\hat{I}_t), \tag{8}$$

where $\text{Laplace}(\cdot)$ denotes the Laplace operator used to extract the second-order gradient of the image, enhancing the response to fine image details.

Ultimately, the reward measure $r(s_t, a_t)$ is defined as the weighted sum of all rewards:

$$r = \lambda_1 r_w + \lambda_2 r_s + \lambda_3 r_l, \tag{9}$$

where, the weights $\lambda_1$, $\lambda_2$, and $\lambda_3$ are chosen empirically to maintain a balanced scale across the three rewards.

## 3.3 Network Structure and Training Strategy

The architectural configuration of the BCSCN is illustrated in Figure 2, employing an Actor-Critic framework comprising the Actor network, Critic network, and neural renderer. The specific structure is detailed as follows: Both the Actor and Critic networks adopt a residual structure similar to ResNet-18 [8]. The Actor network is responsible for generating basis function parameters based on the current state $s_t$, while the Critic network provides an evaluation score based on the current state $s_t$, the current reconstructed image $\hat{I}_t$, and the next-step reconstructed image $\hat{I}_{t+1}$. The differentiable neural renderer employs a 4-layer fully connected network and two layers of convolutional networks to map the parameters of the basis functions to rasterized curve basis, which are then overlaid with $\hat{I}_t$, to generate the next-step reconstructed image $\hat{I}_{t+1}$. During

the training of the BCSCN, the Deep Deterministic Policy Gradient (DDPG) [16] training strategy is employed. The parameters of the differentiable neural renderer are fixed, while the remaining components undergo joint training.

In addition, due to the rich structural information in natural images, accurately reconstructing images at arbitrary scales poses a significant challenge. Leveraging non-local priors inherent in natural image, local patches from different positions exhibit repetitive patterns. Similar to most SR methods which divide the entire image to small patches for training, in our BCSCN training and inference processes, we decompose the input image into consecutive small patches for individual reconstruction, thereby mitigating the difficulty of basis search and achieving more precise reconstruction.

## 4 EXPERIMENTS

To assess the impact of our proposed approach on improving the generalization of SR networks, we integrated BCSCN with SR network, resulting in a new SR network called BCSCN-SR. The training of BCSCN-SR was conducted in multiple stages. Initially, BCSCN underwent training using the reward function specified in Eq 9. Once this stage was completed, the subsequent SR network could be trained based on the preprocessed data.

Three experiments were devised to assess the effectiveness of the proposed method: 1) assessing the impact of BCSCN on domain gap narrowing, and 2) validating the generalization of BCSCN-SR, and 3) to compare it with blind super-resolution approaches.

## 4.1 Dataset

**Training Set:** We utilized the DIV2K dataset [28] to train BCSCN, which comprises 900 HR images. Among these, 800 images were allocated for training, and 100 for validation. The training set was composed of LR images generated by performing bicubic downsampling on the 800 HR images from the DIV2K dataset.

**Testing Set:** We employed four Real World Super-Resolution Challenge (RWSRC) datasets provided by NTIRE and AIM to assess our method. These datasets, namely Ntire2018 Track2 [30], Ntire2018 Track4 [30], Aim2019 Track2 [20], and Ntire2020 Track1 [19], were obtained by downsampling (×4) the HR images from the DIV2K validation set using four distinct methods. The degradation operators used for downsampling are diverse and undisclosed to simulate real-world degradation scenarios, as detailed in the respective papers. Additionally, we incorporated the RealSRSet dataset by Kai et al. [41], featuring 20 real LR images without HR counterparts.

## 4.2 Training Details

To train BCSCN, We randomly cropped patches of size 16×16 from LR samples. The training batch size was set to 96. The agent's replay buffer size was configured as 16,000, allowing a maximum prediction of 20 time steps. The learning rate was initialized to $1 \times 10^{-4}$. After BCSCN training, we utilized it to preprocess LR images, producing "clean" images for subsequent SR training.

## 4.3 Metrics

For the BCSCN preprocessing network, we utilized SIFID [27] to assess its capability in narrowing the distribution discrepancies of degraded images. SIFID has been proven sensitive to variations in

**Table 1: Examining BCSCN's impact on LR image distribution discrepancies between training and testing sets.**

|  | Ntire2018 Track2 | Ntire2018 Track4 | Aim2019 Track2 | Ntire2020 Track1 |
|---|---|---|---|---|
| w/o BCSCN | 7.9097 | 11.9371 | 4.2665 | 1.6521 |
| w BCSCN | **2.6822** | **5.9295** | **1.9743** | **0.0765** |

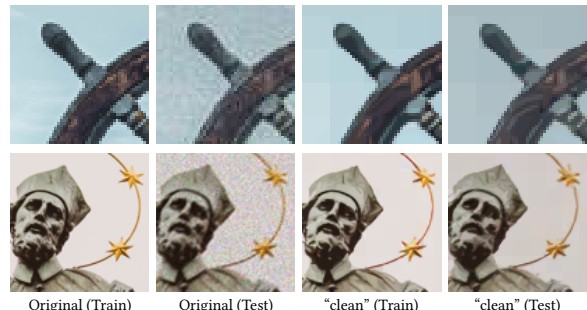

Original (Train)    Original (Test)    "clean" (Train)    "clean" (Test)

**Figure 3: Illustrates two LR images from the training and testing sets, both with and without the application of BCSCN. The term "clean" denotes images processed through BCSCN.**

degradation kernels [25]. To evaluate the performance of BCSCN-SR, we employed PSNR and SSIM [35] evaluation metrics on the Y-channel. Additionally, we incorporated perceptual metrics, including LPIPS [42] based on the AlexNet [13] to gauge the perceptual similarity on the RGB channels, and FID [9] was employed to quantify the distance of distributions between the original and the reconstructed images. For RealSRSet, where the ground truth is not available, we employed no-reference image quality measures oriented towards perceptual evaluation, specifically NIQE [42] and MUSIQ [11] for evaluation. Human perception plays a pivotal role in assessing the effectiveness of super-resolution techniques. In line with previous studies [2, 32, 33], we prioritize perceptual metrics, regarding PSNR and SSIM metrics solely as references.

### 4.4 Experiments of Domain Gap Narrowing

To assess BCSCN's capability in narrowing domain gaps, we measured the disparities in LR image distributions between the training and testing sets, both with and without the application of BCSCN. Specifically, we quantified the dissimilarity in the LR image distribution between the training set and each testing set. Additional, we examined distribution disparities between "clean" images from the training set and each test set. These images undergo preprocessing by BCSCN before being input into SR network of BCSCN-SR.

Table 1 presents the differences in SIFID with and without the BCSCN. In comparison to the disparity observed without BCSCN processing, the distributional gap of "clean" images has significantly diminished, resulting in lower SIFID scores. Figure 3 presents qualitative outcomes, showcasing that the reconstructed "clean" images not only retain essential content and intricate details but also effectively eliminate residual structures, noise, and compression artifacts. These findings show BCSCN significantly reduces the distributional disparities between training and testing images.

### 4.5 Experiments of Generalization

To assess the effectiveness of our method, we introduce two distinct implementations of the BCSCN-SR: BCSCN-ESRGAN and BCSCN-PASD$_b$. BCSCN-ESRGAN combines BCSCN with the classical GAN-based SR method ESRGAN [34]. Considering the potent generative capabilities of diffusion models, we also present a version utilizing diffusion techniques in the hope of achieving enhanced performance. BCSCN-PASD$_b$ integrates BCSCN with the Pixel-Aware Stable Diffusion (PASD) model [40], an SR approach based on stable diffusion (SD) [26]. We solely adopted its network architecture, omitting the degradation supervision component, and fine-tuned the model starting from SD. For training the SR network, we kept the parameters of BCSCN fixed and used a paired LR-HR dataset synthesized through bicubic degradation from 3,000 HR images in the DF2K (DIV2K[28] + Flickr2K [29]) dataset as the training set.

To validate the BCSCN's ability to enhance the generalization performance of SISR models, we carried out comparative assessments against the following SOTA SR methods that similarly were trained using bicubic downsampling: EDSR [17], RCAN [43], NLSA [23] and SwinIR [15]. EDSR represents a classical residual learning-based approach. RCAN, NLSA, and SwinIR employ techniques based on channel attention, non-local attention, and the Swin Transformer, respectively. We also included the ESRGAN and PASD models trained on bicubic downsampling as our baseline models. Except for PASD's bicubic version, PASD$_b$, which required additional training, all baseline models utilized their provided pretrained weights. All methods were trained solely on a training set with a bicubic degradation kernel and then directly applied to test sets with unknown degradation kernels.

#### 4.5.1 Results on Real-World Super-Resolution Challenge Datasets.

Table 2 presents the quantitative evaluation results, demonstrating that the BCSCN preprocessing network significantly enhances the generalization performance of both ESRGAN and PASD$_b$. Specifically, (1) BCSCN-ESRGAN outperforms ESRGAN in LPIPS and FID metrics by 38.0% and 30.5% respectively, while BCSCN-PASD$_b$ surpasses PASD$_b$ by 32.2% and 29.4% in the same metrics, affirming the efficacy of BCSCN in boosting the generalizability of SR models. (2) BCSCN-PASD$_b$ exhibits further advancements in generalization performance compared to BCSCN-ESRGAN, with improvements of 7.2% in LPIPS and 17.8% in FID. This improvement can be attributed to the rich natural image priors within the stable diffusion model, which provide extensive texture details, albeit at the cost of a slight reduction in pixel consistency compared to BCSCN-ESRGAN. (3) Both BCSCN-PASD$_b$ and BCSCN-ESRGAN significantly outperform other baseline methods in terms of overall performance, achieving superior perceptual quality. Furthermore, within the AIM2019Track2 and NTire2020Track1, PSNR and SSIM are not as advantageous, primarily because GAN or diffusion-based approaches can generate more realistic details, albeit with a compromise on pixel consistency.

Figure 4 showcases the qualitative results. The image '0821' from Ntire2018Track2 suffers from severe blurring and noise, making it a challenging task for most baseline methods to effectively restore such images. This is particularly true for ESRGAN, which tends to exacerbate noise, leading to severe artifacts. In contrast, our approach, whether BCSCN-ESRGAN or BCSCN-PASD$_b$, produces

**Table 2: The comparisons between different methods. Best and second best results are highlighted in red and blue, respectively.**

| Dataset | Metrics | EDSR | RCAN | NLSA | SwinIR | ESRGAN | PASD$_b$ | BCSCN-ESRGAN | BCSCN-PASD$_b$ |
|---|---|---|---|---|---|---|---|---|---|
| **NTIRE2018 Track2** | PSNR↑ | 20.4184 | 20.4266 | 20.4195 | 20.4240 | 19.6370 | 20.3295 | 20.5004 | 19.6371 |
| | SSIM↑ | 0.5039 | 0.5043 | 0.5038 | 0.5038 | 0.3834 | 0.4879 | 0.5388 | 0.4613 |
| | LPIPS↓ | 0.7746 | 0.7786 | 0.7800 | 0.7770 | 0.7178 | 0.6569 | 0.5154 | 0.4361 |
| | FID↓ | 93.1343 | 92.2650 | 91.2088 | 92.2883 | 101.9349 | 88.4089 | 74.6498 | 57.4734 |
| **NTIRE2018 Track4** | PSNR↑ | 20.0604 | 20.0624 | 20.0617 | 20.0649 | 19.2322 | 20.0245 | 20.2135 | 19.7570 |
| | SSIM↑ | 0.4762 | 0.4758 | 0.4757 | 0.4758 | 0.3381 | 0.4683 | 0.5172 | 0.4575 |
| | LPIPS↓ | 0.7812 | 0.7847 | 0.7862 | 0.7837 | 0.7510 | 0.6756 | 0.5526 | 0.4714 |
| | FID↓ | 106.3097 | 105.5669 | 104.2503 | 105.7234 | 114.9969 | 102.2008 | 89.1935 | 66.4804 |
| **AIM2019 Track2** | PSNR↑ | 24.1765 | 24.1362 | 24.1984 | 24.1645 | 23.1585 | 23.1956 | 23.4788 | 22.2986 |
| | SSIM↑ | 0.6759 | 0.6749 | 0.6773 | 0.6751 | 0.6180 | 0.6233 | 0.6606 | 0.6014 |
| | LPIPS↓ | 0.6117 | 0.6749 | 0.6134 | 0.6176 | 0.5465 | 0.5044 | 0.3486 | 0.3609 |
| | FID↓ | 85.2881 | 84.6636 | 83.4817 | 85.7457 | 101.3222 | 76.0358 | 56.5406 | 51.2990 |
| **NTIRE2020 Track1** | PSNR↑ | 27.8163 | 27.8621 | 27.7437 | 27.9115 | 21.1277 | 25.8015 | 25.6632 | 23.9412 |
| | SSIM↑ | 0.7341 | 0.7388 | 0.7319 | 0.7390 | 0.3116 | 0.6443 | 0.6950 | 0.6257 |
| | LPIPS↓ | 0.5863 | 0.5833 | 0.5920 | 0.5723 | 0.7588 | 0.4234 | 0.3029 | 0.3282 |
| | FID↓ | 55.9301 | 54.3719 | 54.6026 | 53.7646 | 69.8056 | 56.0860 | 49.4735 | 46.4809 |
| **RealSRSet** | NIQE ↓ | 5.5256 | 5.6101 | 5.7030 | 5.4998 | 4.5883 | 4.4952 | 3.6746 | 3.6240 |
| | MUSIQ ↑ | 45.3460 | 46.0816 | 44.9063 | 45.3077 | 48.9332 | 49.5852 | 57.7467 | 64.5600 |

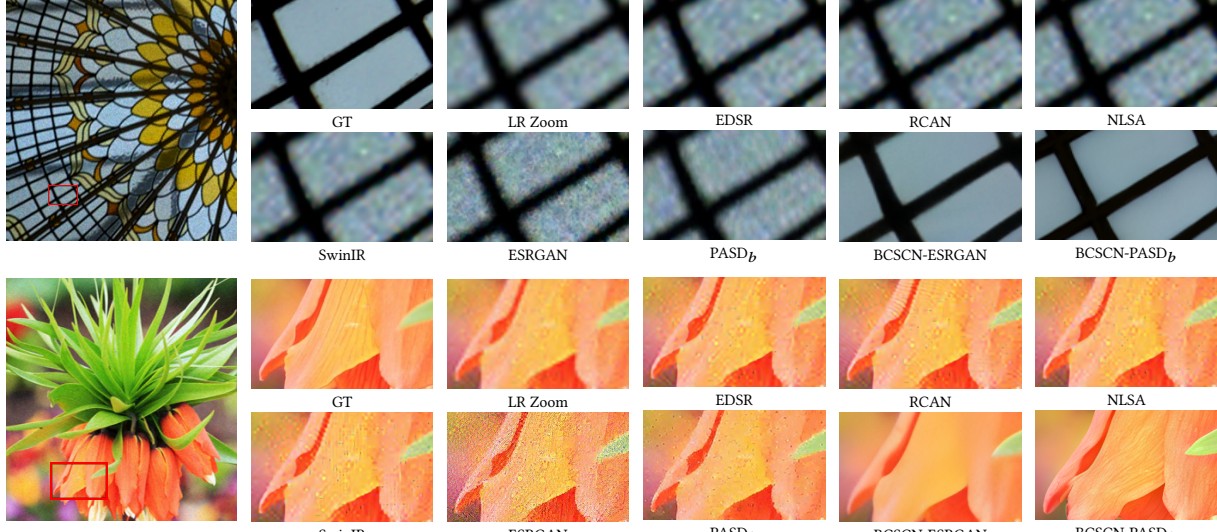

**Figure 4: Visualization results of Real-Word Super-Resolution Challenge Datasets. The first line originates from the '0821' of Ntire2018Track2, and the second line is derived from the '0803' of Ntire2020Track1. More results can be found in the Appendix.**

images with clear and sharp edges. Moreover, while baseline methods can partially restore the edges of the slightly degraded image '0803', they inevitably introduce intolerable artifacts. However, our methods are capable of generating clean and sharp images, with BCSCN-PASD$_b$ being notably effective in restoring the edges of petals and generating surprisingly realistic details.

*4.5.2 Results on RealSRSet Dataset.* Similar to the real-world super-resolution challenge datasets, as illustrated in Table 2, our method achieved the highest NIQE and MUSIQ scores on the RealSRSet,

with BCSCN-PASD$_b$ surpassing the top-ranked baseline method, PASD$_b$, by 19.4% and 30.2% in NIQE and MUSIQ, respectively. Qualitatively, as shown in Figure 5, unlike baseline methods introduce significant high-frequency artifacts, our method maintains robustness against real degradation, restoring clear and clean edges.

## 4.6 Compared with Blind SR Method

In addressing the real-world SR task, besides the method proposed in this paper, there is another popular category of methods known

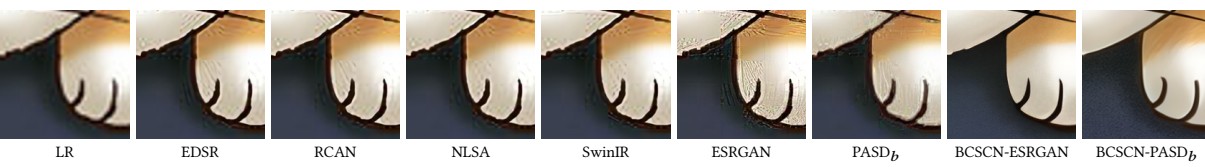

**Figure 5: Visualization results of RealSRSet datasets.**

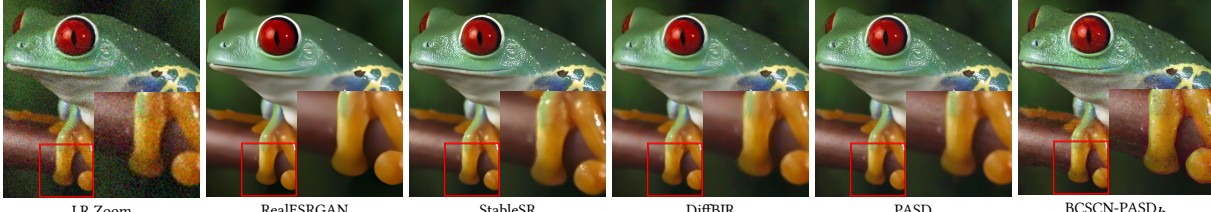

**Figure 6: Qualitative comparisons of various blind super-resolution methods using example from the RealSRSet dataset. More results can be found in the Appendix.**

as blind SR methods. Among these methods, many of the best-performing ones commonly adopt second-order degradation kernels. The term second-order refers to the image degradation process being divided into two consecutive stages. In each stage, a classic degradation model with adjustable parameters is randomly selected to degrade the image. This type of degradation model aims to cover a wide range of degradation scenarios encountered in real-world applications by extensively sampling various combinations of various types and parameters of classic degradation models as much as possible. While these methods, such as RealESRGAN [33], and diffusion-based methods like StableSR [31], DiffBIR [18], and PASD [40], exhibit excellent super-resolution capabilities on artificially degraded data, their performance tends to diminish when applied to real-world scenes where human knowledge of the degradation kernel is lacking. This can be observed from Table 3, where their performance leads on the artificially synthesized datasets RWSRC, but on the real-world dataset RealSRSet, their performance is not as outstanding. The quantitative results of RWSRC in the table are the average across the four RWSRC datasets.

Compared to SOTA blind SR methods, the approach presented in this paper strives to minimize distribution discrepancies and enhance the generalization capability of subsequent SR networks. This is achieved through the proposed preprocessing network BCSCN, which is trained solely on a single, simple bicubic kernel without incorporating any additional degradation. This makes the method less sensitive to changes in the degradation model, and it achieves satisfactory results on both artificially synthesized datasets RWSRC and real-world dataset RealSRSet. Notably, on the Real-SRSet, BCSCN-PASD$_b$ achieves first rank in NIQE and second in MUSIQ, as evidenced by Figure 6, where its ability to recover clear frog legs and branches alongside realistic textures results.

Furthermore, it should be pointed out that, compared to these leading blind super-resolution methods, BCSCN-PASD$_b$ has a higher sample utilization rate and requires fewer resources during training.

For training, RealESRGAN, StableSR, and PASD utilized between 13K to 25K HR images from various datasets, whereas DiffBIR was

**Table 3: The comparisons between blind SR methods.**

| Dataset | Metrics | Real-ESRGAN | StableSR | DiffBIR | PASD | BCSCN-PASD$_b$ |
|---|---|---|---|---|---|---|
| **Degradation Type** | | Second-Order | Second-Order | Second-Order | Second-Order | Bicubic |
| RWSRC | PSNR↑ | 21.900 | 23.1798 | 22.1162 | 22.0582 | 21.4085 |
| | SSIM↑ | 0.6311 | 0.6517 | 0.5234 | 0.5812 | 0.5365 |
| | LPIPS↓ | 0.3384 | 0.3547 | 0.4442 | 0.3417 | 0.3991 |
| | FID↓ | 50.2091 | 42.3528 | 57.1798 | 43.8248 | 55.4334 |
| RealSRSet | NIQE ↓ | 4.2103 | 4.5333 | 3.9272 | 3.7101 | 3.6240 |
| | MUSIQ ↑ | 63.0281 | 59.4793 | 55.2776 | 69.6276 | 64.5600 |

trained on even larger scale samples from the ImageNet [4]. In contrast, BCSCN-PASD$_b$ required only 3K HR images from the DF2K dataset, demonstrating superior sample efficiency.

Moreover, for BCSCN-PASD$_b$, the amount of data in one training epoch is equivalent to the size of the training dataset. In contrast, methods based on second-order degradation kernels process an amount of data per epoch that is equal to the training dataset size multiplied by the number of degradation kernels. This implies that blind SR methods enumerating degradation kernels often require lengthy training periods to converge. Specifically, BCSCN-PASD$_b$ required only 3K HR images and a bicubic degradation model for dataset synthesis. It was trained on four A6000 GPUs for 16 hours, starting from SD pretrained weights, which is only about 10% of the training duration required for PASD.

### 4.7 Ablation

To evaluate the impact of various components of BCSCN on the performance of BCSCN-SR, we build upon the foundation of BCSCN-ESRGAN to conduct a comprehensive analysis across four dimensions: patch size, number of bases, bézier curve basis order, and reward functionality. The overall performance was assessed using average metrics computed across four RWSRC datasets.

*4.7.1 Patch Size and Sparsity Analysis.* Table 4 illustrates the impact of reconstructed patch size on SR outcomes. Adopting a relatively redundant configuration, each scale's patch is systematically

**Table 4: Ablation experiment of patch size.**

| Patch Size | PSNR↑ | SSIM↑ | LPIPS↓ | FID↓ |
|---|---|---|---|---|
| 8×8 | 22.57 | 0.6016 | 0.4406 | 68.02 |
| 16×16 | 22.53 | 0.6013 | 0.4228 | 67.29 |
| 32×32 | 22.06 | 0.5824 | 0.4407 | 72.88 |
| 64×64 | 20.87 | 0.5440 | 0.5035 | 107.31 |

**Table 5: Ablation experiment of basis number.**

| Basis Number | PSNR↑ | SSIM↑ | LPIPS↓ | FID↓ |
|---|---|---|---|---|
| 5 | 21.72 | 0.5788 | 0.4563 | 83.98 |
| 10 | 22.15 | 0.5915 | 0.4315 | 71.46 |
| 20 | 22.46 | 0.6029 | 0.4299 | 67.46 |
| 30 | 22.51 | 0.6013 | 0.4231 | 67.32 |
| 50 | 22.53 | 0.6013 | 0.4228 | 67.29 |

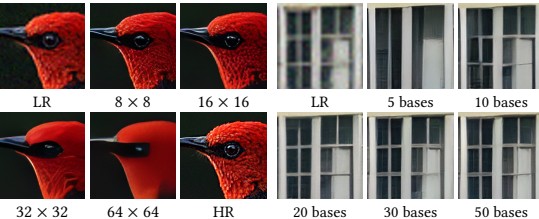

**Figure 7: Comparison of the impact of patch size and number of basis on super-resolution results.**

decomposed into 50 bases. Results reveal a diminishing trend in indicators with increasing patch size. Figure 7 visually demonstrates that larger patch sizes amplify image complexity, challenging the agent in reconstructing intricate details and resulting in smoother SR outcomes. Performance convergence is observed when the patch size falls below 16, with nuances in detail becoming less conspicuous.

Table 5 elucidates the influence of the number of bases on SR results. Maintaining a fixed patch size decomposition at $16 \times 16$, we conduct a comparative analysis for four distinct base quantities, ranging from low to high. Quantitative metrics show swift improvement when the base quantity is below 20, plateauing beyond 20 bases. Figure 7 qualitatively supports this observation, with incremental basis enhancing window pattern intricacies. The transition from 5 to 20 bases manifests a notable escalation in detail, while minimal variation is observed from 20 to 30 bases. The detailed process of decomposing LR images is available in the Appendix.

*4.7.2 Bézier Curve Basis Order Analysis.* Table 6 demonstrates the impact of utilizing Bézier curve bases of different orders on performance. As the order of the bases increases, there is no significant change in model performance, indicating that due to the lower complexity of structures within small image patches, a second-order Bézier curve is sufficient for reconstructing the LR images.

*4.7.3 Reward Analysis.* Table 7 displays the impact of different reward functions on the model's generalization performance. As the number of reward functions increases, LPIPS and FID scores

**Table 6: Ablation experiment of bézier curve basis order.**

| Basis Order | PSNR↑ | SSIM↑ | LPIPS↓ | FID↓ |
|---|---|---|---|---|
| 2 | 22.46 | 0.6029 | 0.4299 | 67.46 |
| 3 | 22.42 | 0.6013 | 0.4275 | 69.21 |
| 4 | 22.30 | 0.5963 | 0.4192 | 67.59 |

improve, affirming each function's effectiveness. Figure 8 shows that when guiding the sparse encoding agent solely with the WGAN reward, clean edge structures are restored but introduce artifacts. As the regional reward is enhanced, these artifacts diminish, albeit with a reduction in image texture details. However, by introducing additional Laplace rewards, lost details are effectively recovered, resulting in a high-quality SR output that is both clean and sharp.

**Table 7: Ablation experiment of reward function.**

| $r_w$ | $r_s$ | $r_l$ | PSNR↑ | SSIM↑ | LPIPS↓ | FID↓ |
|---|---|---|---|---|---|---|
| ✓ | | | 22.50 | 0.5791 | 0.4605 | 72.31 |
| ✓ | ✓ | | 22.28 | 0.5980 | 0.4345 | 70.47 |
| ✓ | ✓ | ✓ | 22.46 | 0.6029 | 0.4299 | 67.46 |

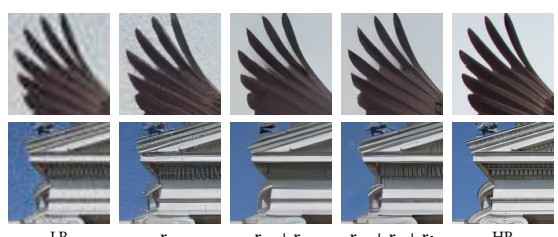

**Figure 8: Visualization results of reward ablation experiment.**

## 5 CONCLUSION

In this study, we improve the generalization of SISR by narrowing the domain gap between training and testing sets. We propose BCSCN, a Bézier curve basis-based sparse coding preprocessing network that effectively addresses distribution disparities in both training and testing phases. BCSCN preserves image content while eliminating degradation-related defects, producing "clean" images for subsequent SR networks. With minimal training on LR images created by bicubic degradation, BCSCN can effectively reduce the distribution distance between varying degradation data, significantly boosting the generalization capability of foundational SR networks and surpassing SISR methods trained with identical degradation kernels. Compared to SOTA blind SR methods, BCSCN-$\text{PASD}_b$ is trained on simple degradations, utilizing fewer samples and training resources, yet achieves competitive performance on RWSC datasets and securing the highest NIQE and the second-highest MUSIQ scores on the RealSRSet. These results demonstrate BCSCN's significant role in improving the generalization performance of base SISR models, affirming its effective application in real-world scenarios.

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
