# OpenReview forum: "BCSCN:Reducing Domain Gap through Bézier Curve basis-based Sparse Coding Network for Single-Image Super-Resolution"
_acmmm.org/ACMMM/2024/Conference — MM2024 Poster_

### Official Review · Reviewer_xLMV · 2024-05-05

**Rating:** 5
**Confidence:** 2

**Summary:**

In summary, this article presents BCSCN, a preprocessing network based on the Bézier curve basis, effectively addressing domain gap challenges by minimizing input distribution discrepancies between training and testing phases. The article introduces a set of rewards to guide the basis coefficients search agent in capturing main content and eliminating degradation-related information. Integration of BCSCN into GAN-based and diffusion-based SR networks, resulting in BCSCN-ESRGAN and BCSCN-PASD𝑏, respectively, yields significant improvements in naturalness performance on real-world datasets with unknown degradation kernels compared to their base models. Moreover, BCSCN-PASD𝑏 achieves competitive results to state-of-the-art blind SR methods while requiring fewer samples and lower training resources.

**Strengths:**

This article introduces several noteworthy contributions. Firstly, BCSCN, a preprocessing network utilizing the Bézier curve basis, effectively reduces input distribution discrepancies between training and testing phases, addressing the domain gap challenge in subsequent SR networks. Secondly, the article presents a set of rewards aimed at guiding the basis coefficients search agent to capture main content and eliminate degradation-related information. Thirdly, the integration of BCSCN into GAN-based and diffusion-based SR networks, resulting in BCSCN-ESRGAN and BCSCN-PASD𝑏 respectively, leads to significant enhancements in naturalness performance on real-world datasets with unknown degradation kernels compared to their base models. Moreover, BCSCN-PASD𝑏 demonstrates competitive results against state-of-the-art blind SR methods while requiring fewer samples and lower training resources.

**Limitations:**

While the proposed BCSCN effectively addresses distribution disparities in both training and testing phases, there are certain aspects that warrant clarification. In the ablation study on different patch sizes (Section 4.7), the authors suggest that larger patch sizes amplify image complexity, challenging the reconstruction of intricate details and resulting in smoother SR outcomes. This raises two important questions: 1) How would the performance be affected if a patch size of 4x4 is used? 2) Following the authors' reasoning, could increasing the Basis Number mitigate the challenges posed by larger patch sizes? If not, are there alternative explanations for this phenomenon?

**Suitability:**

2

---

### Official Review · Reviewer_TRPo · 2024-05-08

**Rating:** 2
**Confidence:** 3

**Summary:**

The authors propose the Bézier Curve basis-based Sparse Coding Network (BCSCN), which is a preprocessing network for extracting clean subjects from input images containing various types of degradation. BCSCN has adopted a reinforcement learning framework and three different types of reward function to produce the optimal basis coefficients within a predefined Bézier Curve basis space.

**Strengths:**

1.The authors propose a Bézier Curve basis-based Sparse Coding Network. The network skillfully combines a sparse coding framework with reinforcement learning in the preprocessing step, while keeping the super-resolution model unchanged.
2.The authors designed a reward function with three components, focusing on the trade-off between the reconstruction distance of WGAN, a larger curve area and rich details.
3.The experimental results indicates BCSCN could improve the current SOTA models.

**Limitations:**

1.The authors decompose the input image into consecutive small patches for individual reconstruction but do not employ any post-processing mechanism to maintain consistency across the image as in other traditional sparse coded super-resolution models. The inconsistency is evident in Figure 3.
2.Insufficient experimentation. The four RWSRC datasets used were from the same 100 images of natural scenes, only with different degradation. The RealSRSet dataset is only 20 images. It is unable to show the model's ability to process other types of images.
3.The number of BCSCN iterations and the additional inference time due to BCSCN are not shown in the experiments.

**Suitability:**

2

---

### Official Review · Reviewer_6PBQ · 2024-05-20

**Rating:** 4
**Confidence:** 3

**Summary:**

This paper aims at narrowing the domain gap between training and testing sets, thus improving the generalization capability of SR models. Specifically, it proposes a Bézier curve basis-based sparse coding preprocessing network BCSCN to solve the distribution disparities between training and testing phases. The underlying principle is to preserve the image content as well as eliminate degradation-related defects (e.g., artifacts, residual structures, and noise), so as to produce “clean” images for subsequent SR operations.

**Strengths:**

Novelty. Moderate original.

Clarity. Well organized and easy to understand.

Technical Correctness. Probably correct.

Reference to Prior Work. References adequate.

**Limitations:**

Inadequate rationalization of experimental settings. According to the authors, BCSCN can be considered as a pre-processing process for SR restoration models. Therefore, it should be compared with solutions with similar ideas or operations. For example, CinCGAN [1] and MCinCGAN [2] can be regarded as representative two-stage restoration algorithms, where the first stage is mainly aimed at obtaining clean LR images, while the second stage is mainly responsible for SR reconstruction of clear LR images. This manuscript should take such studies into account.

[1] Yuan Y, Liu S, Zhang J, et al. Unsupervised image super-resolution using cycle-in-cycle generative adversarial networks[C]//Proceedings of the IEEE conference on computer vision and pattern recognition workshops. 2018: 701-710.

[2] Zhang Y, Liu S, Dong C, et al. Multiple cycle-in-cycle generative adversarial networks for unsupervised image super-resolution[J]. IEEE transactions on Image Processing, 2019, 29: 1101-1112.

Limited experimental validation. The authors used only four real-world SR challenge (RWSRC) datasets to validate the effectiveness of the proposed method. First of all, these datasets are poorly accessible unless the authors can provide their download links. In addition, for more fair comparison, there are already some recognized realistic SR benchmarks, such as RealSR [3], DRealSR [4], DPED [5], etc., which can be used to demonstrate the strong generalization ability of practical SR solutions.

[3] Cai J, Zeng H, Yong H, et al. Toward real-world single image super-resolution: A new benchmark and a new model[C]//Proceedings of the IEEE/CVF international conference on computer vision. 2019: 3086-3095.

[4] Wei P, Xie Z, Lu H, et al. Component divide-and-conquer for real-world image super-resolution[C]//Computer Vision–ECCV 2020: 16th European Conference, Glasgow, UK, August 23–28, 2020, Proceedings, Part VIII 16. Springer International Publishing, 2020: 101-117.

[5] Ignatov A, Kobyshev N, Timofte R, et al. Dslr-quality photos on mobile devices with deep convolutional networks[C]//Proceedings of the IEEE international conference on computer vision. 2017: 3277-3285.

**Suitability:**

2

---

### Official Review · Reviewer_MEYR · 2024-05-24

**Rating:** 5
**Confidence:** 3

**Summary:**

The paper introduces the Bézier Curve basis-based Sparse Coding Network (BCSCN) to improve Single Image Super-Resolution (SISR) by mitigating domain gaps caused by diverse degradation kernels. BCSCN removes visual defects in low-resolution images and uses rewards to maintain main content while eliminating degradation effects. Experiments show that BCSCN enhances the generalization of super-resolution networks.

**Strengths:**

This paper attempts to achieve Image Super-Resolution using a different approach.

**Limitations:**

The comparison with Bézier Curve Sparse Coding in the paper could be expanded to help readers better understand this concept.

In Table 2, the paper only compares with methods like ESRGAN, SwinIR, and PASD, which are not very recent works. It is suggested to include comparisons with more recent methods and networks.

The paper states, "solely on a single, simple bicubic kernel, this makes the method less sensitive to changes in the degradation model." Why would training only on a bicubic kernel make the method less sensitive to changes in the degradation model? Could this method be trained on second-order degradation types? If so, what would the results look like?

**Suitability:**

2

---

### Meta-Review · Area_Chair_USat · 2024-07-05

**Recommendation:** Accept (Poster)
**Confidence:** 5

**Metareview:**

This paper received mixed scores in pre-rebuttal review round: 1 Weak Reject, 1 Borderline Accept, 2 Weak Accept. Thanks to the nice rebuttal, two reviewers raised scores (from Weak Reject to Borderline Reject, and from Weak Accept to Accept). Finally, it seems all reviewers are postive on this paper, including the reviewer rating Borderline Reject was mostly satisfied with the feedbacks in rebuttal. AC believes that this paper can be accepted, and the authors have to incorporate the core items in rebuttal into the main paper when preparing the camera ready.